# A Faster R-CNN-Based Model for the Identification of Weed Seedling

Ye Mu [1,2,3,4], Ruilong Feng [1], Ruiwen Ni [1], Ji Li [1], Tianye Luo [1], Tonghe Liu [1], Xue Li [1], He Gong [1,2,3,4], Ying Guo [1,2,3,4], Yu Sun [1,2,3,4], Yu Bao [5], Shijun Li [6,7], Yingkai Wang [8] and Tianli Hu [1,2,3,4,*]

1   College of Information Technology, Jilin Agricultural University, Changchun 130118, China
2   Jilin Province Agricultural Internet of Things Technology Collaborative Innovation Center, Changchun 130118, China
3   Jilin Province Intelligent Environmental Engineering Research Center, Changchun 130118, China
4   Jilin Province Colleges and Universities and the 13th Five-Year Engineering Research Center, Changchun 130118, China
5   School of Management, Changchun University, Changchun 130022, China
6   College of Information Technology, Wuzhou University, Wuzhou 543003, China
7   Guangxi Key Laboratory of Machine Vision and Inteligent Control, Wuzhou 543003, China
8   College of Agriculture, Jilin Agricultural University, Changchun 130118, China
*   Correspondence: hutianli@jlau.edu.cn

**Abstract:** The accurate and rapid acquisition of crop and weed information is an important prerequisite for automated weeding operations. This paper proposes the application of a network model based on Faster R-CNN for weed identification in images of cropping areas. The feature pyramid network (FPN) algorithm is integrated into the Faster R-CNN network to improve recognition accuracy. The Faster R-CNN deep learning network model is used to share convolution features, and the ResNeXt network is fused with FPN for feature extractions. Tests using >3000 images for training and >1000 images for testing demonstrate a recognition accuracy of >95%. The proposed method can effectively detect weeds in images with complex backgrounds taken in the field, thereby facilitating accurate automated weed control systems.

**Keywords:** weed identification; Faster-R-CNN; FPN; ResNeXt

## 1. Introduction

*Weeds* include all kinds of herbaceous plants that grow where they are not wanted [1,2]. Weeds cause much harm to the agricultural economy [3]. Not only do they compete with crops for sunlight, water and fertilizer, but they also compete for living space. If not dealt with in time, weeds will reduce crop growth, yield and quality, and can even cause crop failure. With the development of computer technology, rapid and accurate machine vision-based recognition technology has been increasingly used for weed identification [4–7]. Deep learning has achieved good results in the recognition of human behaviour, crop fruits and weeds. The image recognition method of kiwifruit based on a convolutional neural network (CNN) shows that CNNs have good application prospects in field fruit recognition. Recognition of corn weeds based on CNNs, hash codes and multi-scale hierarchical features has proven the effectiveness of CNNs in recognising weeds in field images. In a study on broccoli seedlings, a crop detection method based on the Faster R-CNN model was proposed. Using a dropout value of 0.6, the ResNet101 network was used as the feature extraction network, and an average precision of 91.73% was achieved. At present, the Faster R-CNN model is also widely applied in the field of vehicle detection [8], ground object recognition in remote sensing images [9,10], appearance defect detection [11], pedestrian detection and recognition [12], and field image detection [13], and has excellent recognition accuracy.

Since most current models have the shortcomings of high complexity and difficulty in modification, an improved Faster R-CNN field weed detection model is proposed in this study. This network model was fused with the Feature Pyramid Network (FPN) to improve the detection accuracy of the model in weed identification. On the basis of obtaining a large amount of weed and plant seedling data, a deep learning network is established, and the ResNeXt feature extraction network is used to generate target recognition models for different types of weeds. After training, it can effectively identify weeds and plant seedlings in images. In this way, a Faster R-CNN-based rapid weed identification system for use in the field is obtained, which can identify weeds in multiple crops.

## 2. Materials and Methods

### 2.1. Image Data Acquisition

Image data were obtained from the V2 Plant Seedlings Dataset [14]. This dataset consists of images of nine field weeds—scentless mayweed, common chickweed, shepherd's purse, cleavers, charlock, fat hen, small-flowered cranesbill, black-grass, and loose silky-bent—and three crop seedlings—maize, common wheat, and sugar beet (Figure 1). To ensure experimental accuracy, the images were collected under different conditions, including sunny, cloudy, and rainy days. There were 5539 images in total, including 598 of scentless mayweed, 713 of common chickweed, 274 of shepherd's purse, 335 of cleavers, 452 of charlock, 538 of fat hen, 576 of small-flowered cranesbill, 309 of black-grass, 762 of loose silky-bent, 257 of maize, 253 of common wheat, and 463 of sugar beet. The resolution of the images after processing was $227 \times 227$ pixels. All processed images were divided as follows: 3329 in a training set, 1107 in a validation set, and 1103 in a testing set. The pictures in these sets did not overlap each other. The training set was used to train the model parameters, the test set was used to evaluate the generalization error of the model applied to the samples after training, and the validation set was used to tune the hyperparameters of the model during the training process. After the experiment was completed, deep learning was used to apply evaluation indicators typically used in the field of object recognition to measure the performance of the detector model.

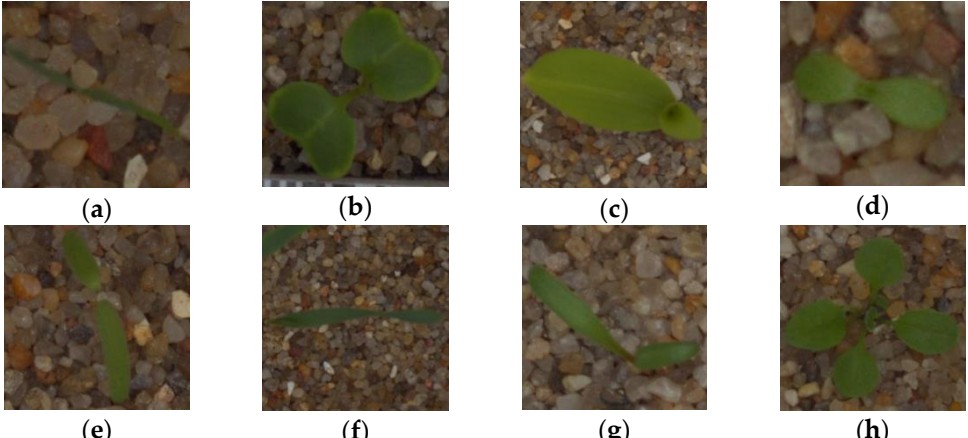

**Figure 1.** Examples of partial samples of (**a**) black-grass; (**b**) charlock; (**c**) maize; (**d**) scentless mayweed; (**e**) fat hen; (**f**) common wheat; (**g**) sugar beet; and (**h**) shepherd's purse.

This dataset is due to the use of public datasets, so there are some limitations. Because the pictures in the dataset exist as a single crop, they cannot be used for large-area crop identification between fields. Moreover, because the dataset collection site is under laboratory conditions, a large amount of sand and gravel is selected for plant cultivation, so the detection accuracy of soil-growing plants cannot be guaranteed.

*2.2. Image Pre-Processing*

Since the images were affected by the environmental conditions at the time of collection, extreme highlights and shadows could decrease the model's recognition success rate. Therefore, the images were processed and segmented; the green part of the plant was extracted and non-green areas were suppressed. The application of normalized colour components can effectively improve the effects of lighting and shadows on image quality [15,16]. After the greyscale images were obtained, the Otsu method [17,18] was used to convert them into binary images and separate the plants. After processing, clear plant images were obtained. They were then filled with noise to obtain single-leaf binary images after segmentation image processing, as shown in Figure 2.

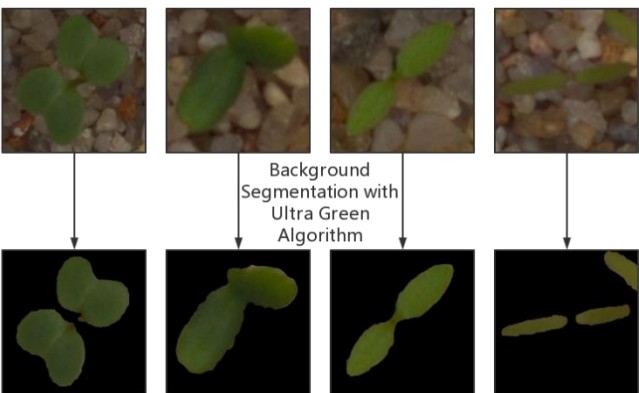

**Figure 2.** Example of single-leaf binary images after partial processing.

## 3. Model Refinement

Currently, there are two types of object detection models. The first is one complete object recognition and object positioning in two steps. Typical representatives of this approach are the R-CNN, Fast R-CNN, and Faster R-CNN families. This type of framework has a low recognition error rate and a low omission rate and can be used in real-time detection scenarios. The typical representative of the second category are YOLO (you only look once), SSD (Single Shot Multibox Detector), YOLOv2, and YOLOv3. To complete object classification and object localization in one step [19–21], SSD directly regresses the position and category of the target at the output layer. Although these methods have fast recognition speed, their accuracy rate is lower than that of Faster R-CNN. Therefore, this paper selected the Faster R-CNN model framework to identify weed images.

The network structure of Faster R-CNN is shown in Figure 3. The network can be roughly divided into four parts: (1) a feature extraction layer, (2) a Region Proposal Network (RPN), (3) a Region of Interest pooling (ROI pooling) layer, and (4) classification and regression.

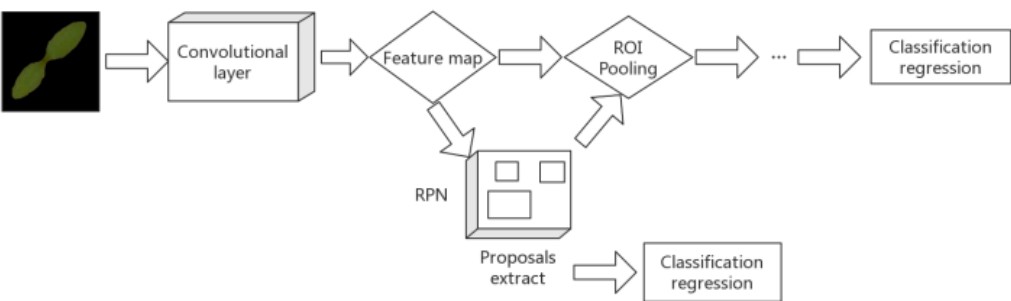

**Figure 3.** The Faster R-CNN network structure.

The deep network in the convolutional network is used to respond to the semantics, and the shallow layer responds to the image. However, in object detection, because the

feature map size is too small, the high-level network can respond to the semantic features but is not conducive to object detection [22]. Therefore, a Feature Pyramid Network (FPN) is introduced to improve the weed detection accuracy of the algorithm in the field identification process. The structure of the FPN network is shown in Figure 4.

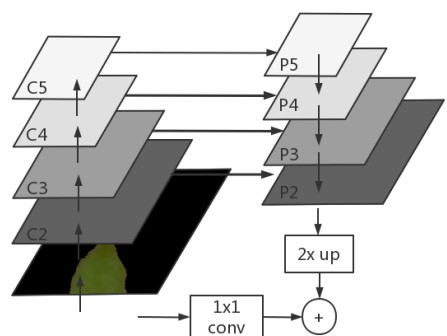

**Figure 4.** The FPN network structure diagram.

Bottom-to-top process: It can be seen that the bottom-to-top extraction process is a conventional feature extraction process. The output of the last layer of each stage is selected as the input of the feature map. The first layer is not used here due to its large memory footprint.

Top-down process: Upsampling starting from the highest layer yields better feature maps, and nearest-neighbour 2× upsampling is used for simplicity.

Horizontal connection process: Each horizontal connection fuses feature maps of the same spatial size, specifically, the upsampling result and the bottom-up C2, C3, C4, and C5 layers are fused with the same-sized feature maps generated by $1 \times 1$ convolution. The output channels are all set to the same 256 channels.

The ResNeXt target extraction network and FPN are selected for fusion in the Faster R-CNN network. This is used to extract target features in target detection so that the learning of the target features is more complete.

The building blocks of ResNet and ResNeXt are shown in Figure 5. While the ResNeXt network structure retains the basic stacking method of ResNet, it splits its single path into 32 independent paths, which perform convolution operations on the input image at the same time. Finally, the cumulative summation of outputs from different paths is used as the final result. This operation makes the division of labour of the network clearer and the local adaptability stronger. Since each path shares the same topology and convolution parameters, and the design method is the same, the network parameters will not increase, which is convenient for model transplantation.

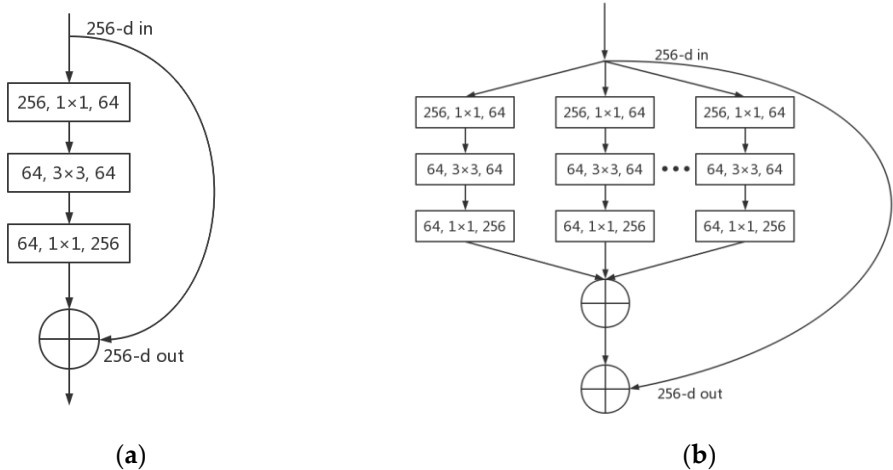

(a)   (b)

**Figure 5.** Building blocks of (**a**) ResNet and (**b**) ResNeXt.

The improved model is shown in Figure 6. At present, there are mainly two training methods for Faster R-CNN: alternating training and approximate joint training. The alternate training method trains two networks—RPN and Fast-RCNN—in a total of two stages, with each stage training the RPN and Fast-RCNN once. In the approximate joint training process, only one weighted network is trained, which requires slightly less memory. This training method saves 25–50% of the training time compared with the alternate training method, and the two training methods have similar accuracy, so the approximate joint training method was selected.

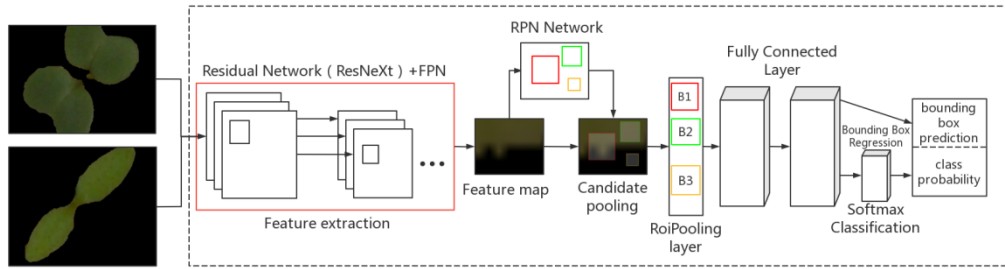

**Figure 6.** Improved model plots.

First, the weed image is input into the model and the improved ResNeXt network is used to fuse the FPN network to extract features from the image. The overall design of FPN is presented in a multi-scale pyramid structure, and each layer of FPN corresponds to {P2, P3, P4, P5, P6} in ResNeXt pyramid through anchors, as shown in Figure 7. Using 3 proportions {1:2, 1:1, 2:1}, 15 types of anchors were used to predict the target objects in weed images in the field. After the feature map is obtained, it is input into the RPN network and propagated forward to obtain a higher-dimensional feature map. When the feature map is passed to the RPN network, the proposed box is obtained and a non-maximum suppression value operation is performed on the proposed box. The top *N* highest-scoring proposal boxes are used as ROIs. The feature map and ROIs are passed to the ROI pooling layer for a pooling operation. Then, the ROI pooling layer is output to the full link layer and linear regression is performed on each area of interest to obtain accurate detection of weeds. Then, the Softmax regression model is used to perform multi-classification target detection and, finally, the categories of weeds and crop seedlings in different fields can be identified.

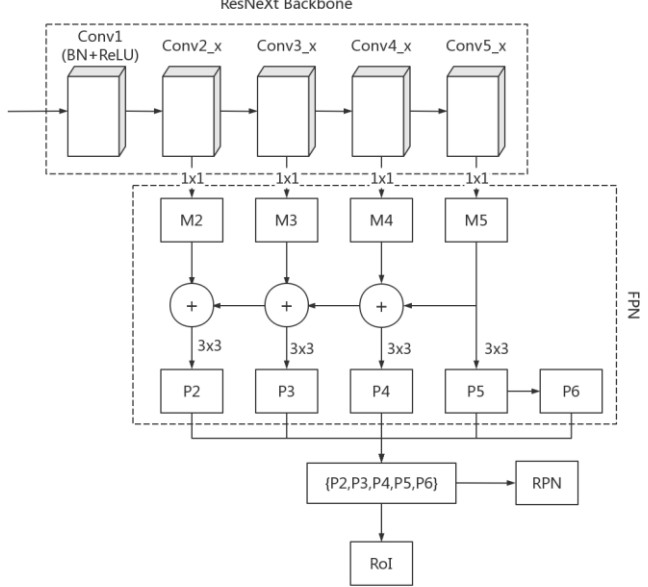

**Figure 7.** Schematic representation of the ResNeXt + FPN structure.

## 4. Experimental Results and Analysis

### 4.1. Test Platform

The operating environment for the test was a desktop computer with a Windows 10 64-bit operating system, and the computer memory was a 32 GB, NVIDIA GeForce RTX 3070 graphics card, equipped with an AMD Ryzen 7 5800H processor.

The software environments used are Anaconda 3.9.12 (Developed by Anaconda, Inc. in Austin, TX, USA), Python 3.7.9, CUDA 11.2 (Developed by NVIDIA Corporation of California, USA), and cuDNN 8.2.1 (Developed by NVIDIA Corporation of California, USA). The open-source deep learning framework Tensorflow 2.0 (Developed by Google, Inc. in California, USA) was used as the development environment.

### 4.2. Parameter Design

To improve the performance of the model and reduce overfitting, the pre-trained model was used to initialize the parameters and the stochastic gradient descent (SGD) method was used to improve the model. The learning rate was set to 0.001, the momentum factor was set to 0.9, the epoch was set to 1500, the maximum number of iterations was 200,000, and the learning rate was adjusted to 0.0001 after 80,000 iterations and to 0.00001 after 160,000 iterations. After the model calculation was completed, the model with the highest accuracy was selected for weed identification. The non-maximum suppression (NMS) value was set to 0.3 to obtain the best candidate box. Finally, to obtain the trained network model, the test set was used to further verify the modelling effect, and the recognition result was the output.

### 4.3. Evaluating Indicator

In this paper, two evaluation indexes, accuracy (*P*) and recall (*R*), were used to verify whether the model can be used for image recognition of weed seedlings. The range of both is [0, 1]. In addition, the $F_1$ value was used to harmonize the average evaluation of the calculation results [23], and the Mean Intersection over Union (MIoU) was used to evaluate the image segmentation results. The assessment is calculated as follows:

$$P = \frac{TP}{TP + FP} \times 100\%, \tag{1}$$

$$R = \frac{TP}{TP + FN} \times 100\%, \tag{2}$$

$$F_1 = \frac{2PR}{P + R} \times 100\%, \tag{3}$$

$$MIoU = \frac{1}{k+1} \sum_{i=0}^{k} \frac{TP}{FN + FP + TP} \tag{4}$$

where *P* stands for accuracy for weed image recognition; *R* is a recall for weed image recognition; $F_1$ is the harmonic mean of *P* and *R*; *TP* is the true positive rate (number of correctly identified crop seedlings and weed targets); *FP* is the number of incorrectly identified crop seedlings and weed targets; *FN* is the number of unidentified crop seedlings and weed targets; *k* is how many categories there are in the dataset.

## 5. Results and Analysis

### 5.1. Model Training Results

According to the above experimental methods, a weed recognition network based on Faster R-CNN fused with FPN was trained and the improved ResNeXt-101 feature extraction network was adopted. The loss degree and accuracy of this model in recognition are shown in Figures 8 and 9.

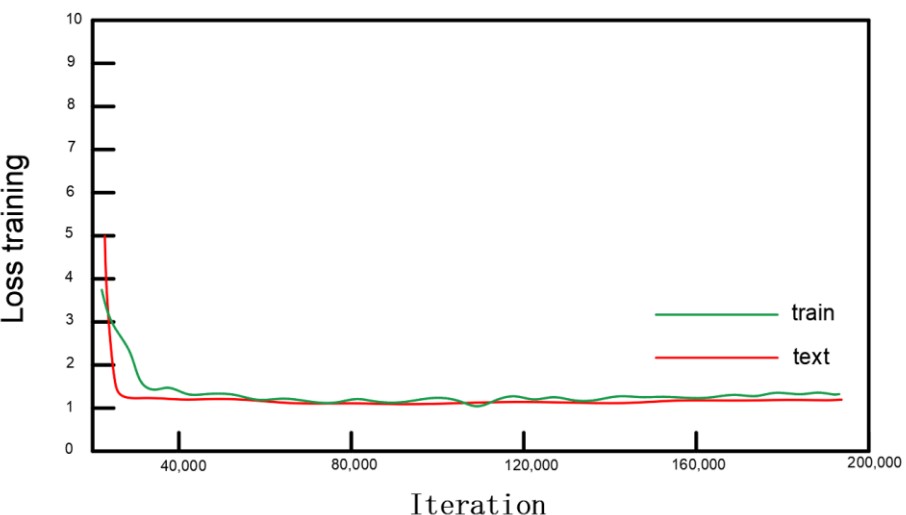

**Figure 8.** Loss of the Faster R-CNN model incorporating FPN.

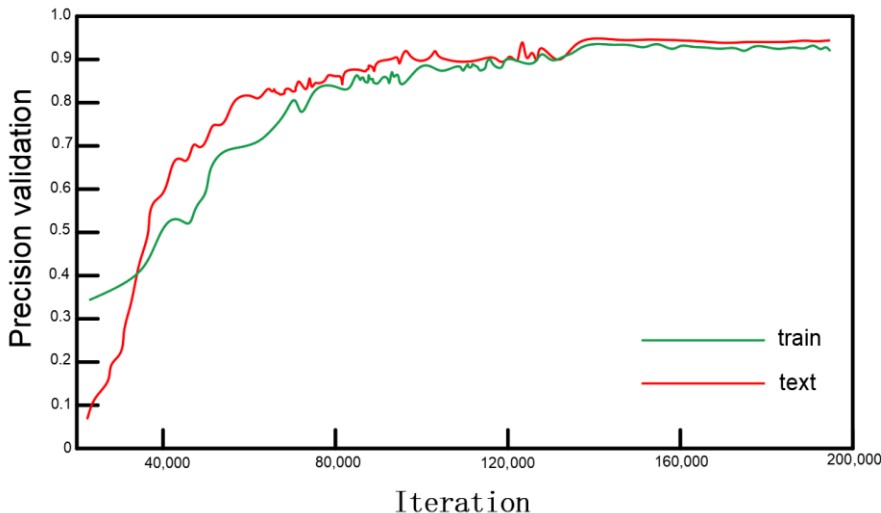

**Figure 9.** Precision of the Faster R-CNN model incorporating FPN.

### 5.2. Experimental Results

It can be seen from Figures 8 and 9 that the weed recognition network based on Faster R-CNN integrated with FPN has high recognition accuracy when using the ResNeXt-101 feature extraction network. To select a better deep network model and a suitable feature extraction network, this experiment compares the Faster R-CNN integrated with FPN and the ordinary Faster R-CNN network under the condition of the ResNeXt-101 feature extraction network. The results are shown in Table 1.

**Table 1.** Performance comparison of the faster R-CNN-FPN network and ordinary Faster R-CNN.

| Model (With ResNeXt-101) | Accuracy (%) | Recall (%) | $F_1$-Value (%) | MIoU (%) | Detection Time (ms) |
|---|---|---|---|---|---|
| Faster R-CNN-FPN | 95.61 | 87.26 | 91.24 | 93.7 | 330 |
| Faster R-CNN | 92.4 | 85.2 | 88.65 | 89.6 | 319 |

It can be seen from Table 1 that Faster R-CNN has a higher accuracy rate when it does not integrate the FPN and only uses the ResNeXt-101 feature extraction network. However, with integrated FPN, the accuracy is higher than that of the ordinary Faster R-CNN network, and the $F_1$-value is higher; in addition, the value of MIOU is about 4% higher than the

unimproved model. It can be seen that the FPN can optimize the network after being integrated into the Faster R-CNN network, which improves the recognition accuracy.

With the Faster R-CNN deep network model integrated with FPN, compared with the ResNet-50 feature extraction network in the literature [22] that proposed FPN, ResNet-50 and ResNeXt-101 were used for training. With increases in the number of model training iterations, the real-time losses of the overall loss functions of the two feature extraction networks used in the Faster R-CNN deep network model were compared (Figure 10 and Table 2).

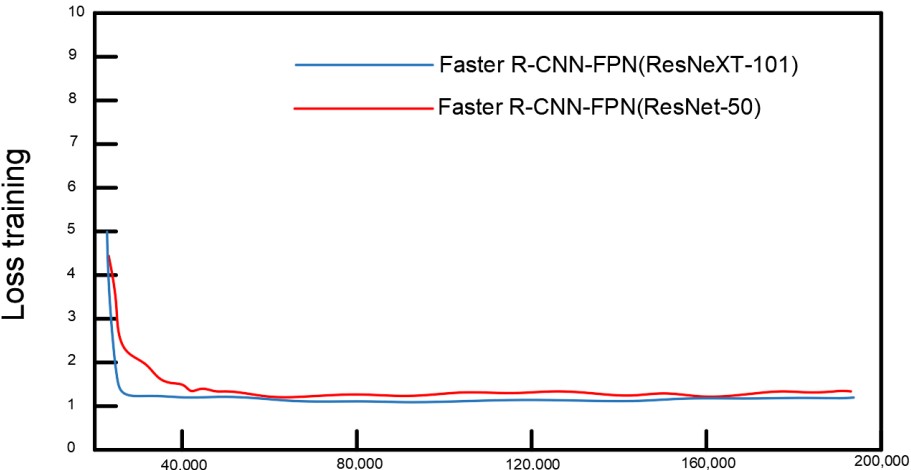

**Figure 10.** Overall loss comparison with the Faster R-CNN model.

**Table 2.** Feature extraction network comparison.

| Model | Extraction Network | Accuracy (%) | Recall (%) | $F_1$-Value (%) | MIoU (%) | Detection Time (ms) |
|---|---|---|---|---|---|---|
| Faster R-CNN-FPN | ResNeXt-101 | 95.61 | 87.26 | 91.24 | 93.7 | 330 |
| Faster R-CNN-FPN | ResNet-50 | 91.67 | 85.19 | 88.31 | 89.5 | 319 |

Figure 10 shows the losses of the overall loss functions of the two feature extraction networks with the Faster R-CNN deep network model. The Faster R-CNN deep network model based on the ResNeXt-101 feature extraction network can converge faster and is obviously better than the ResNet-50 feature extraction network model. The Faster R-CNN deep network model of the ResNet-50 feature extraction network oscillated to varying degrees after 120,000 iterations. The comparison in Table 2 shows that although the ResNeXt-101 feature extraction network uses 101 layers, there is basically no difference between the operation time, but the $F_1$-value and MIoU value of ResNeXt-101 as the result of the feature extraction network are higher than those of the ResNet-50 feature extraction network; therefore, the ResNeXt-101 network has a higher operation accuracy. Since the ResNeXt network replaces the three-layer convolutional blocks of the original ResNet with blocks of the same topology stacked in parallel, it can improve the accuracy of the model without significantly increasing the number of parameters. Compared with ResNet-50, it can be seen that ResNeXt-101 has certain advantages.

To determine the superiority of the deep network model comprised of the Faster R-CNN integrated with the FPN in weed recognition, the results of the three deep network models and the corresponding feature extraction networks were compared. The recognition performance of the ResNeXt-101 feature extraction network was the best. Therefore, on the premise of satisfying the recognition time, ResNeXt-101 was selected as the feature extraction network to compare the overall loss of the Faster R-CNN model, YOLOv3

model, and SSD model, and the accuracy of model verification. The results are shown in Figures 11 and 12 and Table 3.

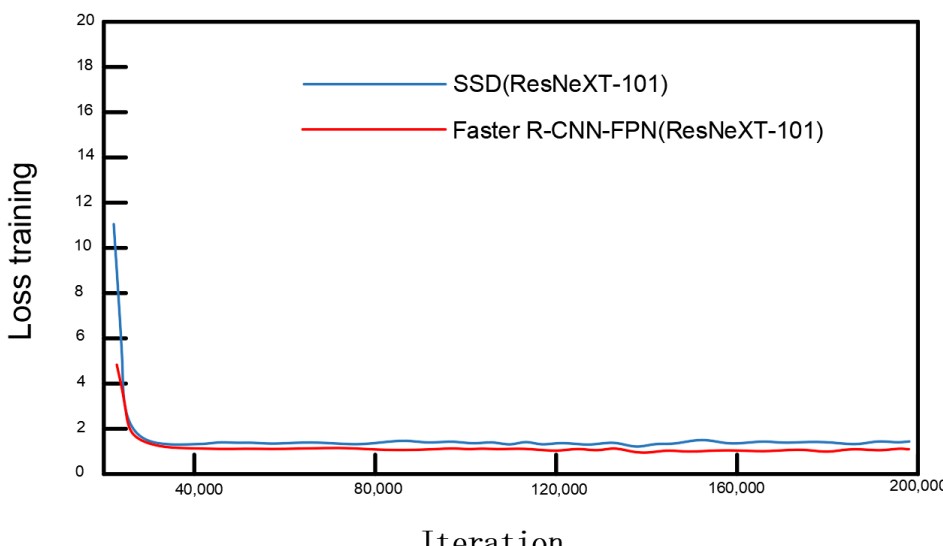

**Figure 11.** Comparison of the overall losses of the Faster R-CNN and SSD models.

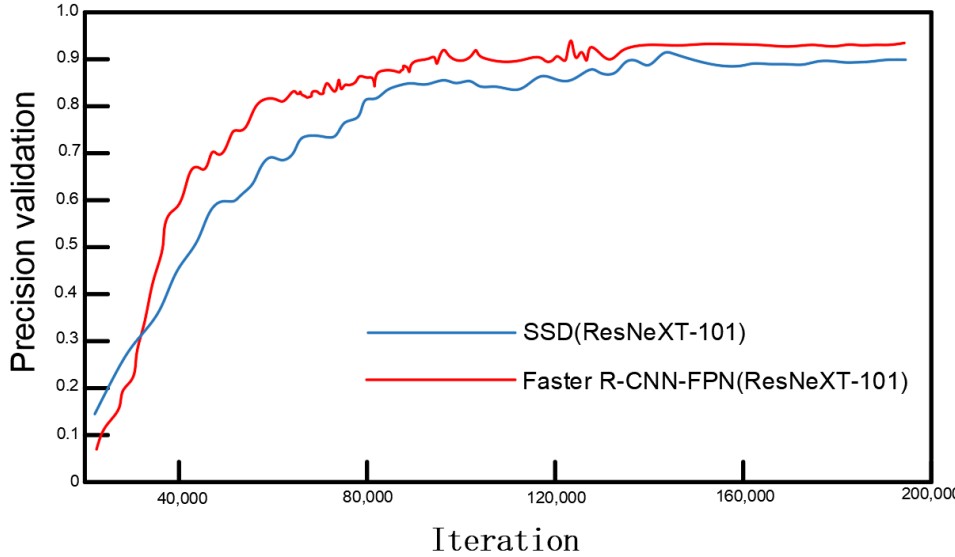

**Figure 12.** Comparison of the precision of the Faster R-CNN and SSD models.

**Table 3.** Deep network model comparison.

| Model | Extraction Network | Accuracy (%) | Recall (%) | $F_1$-Value (%) | MIoU (%) | Detection Time (ms) |
|---|---|---|---|---|---|---|
| Faster R-CNN-FPN | ResNeXt-101 | 95.61 | 87.26 | 91.24 | 93.7 | 330 |
| YOLOv3 | ResNeXt-101 | 84.34 | 79.85 | 82.03 | 84.5 | 215 |
| SSD | ResNeXt-101 | 88.12 | 79.69 | 83.69 | 85.3 | 203 |

The Faster R-CNN, SSD, and YOLOv3 deep network models are compared under the same image processing method and RESNEXT-101 feature extraction network. It can be seen from Figures 11 and 12 that when using the ResNeXt-101 feature extraction network, the SSD network has a slight oscillation at 150,000 steps, while the Faster R-CNN network

tends to stabilize after 120,000 steps. The network loss curve of Faster R-CNN with FPN is lower than that of the SSD deep network model after 4000 iterations, and the network model of Faster R-CNN integrated with FPN is better than the SSD model in terms of loss degree. By comparing the accuracy, recall rate and F-value, it can be seen that, compared with the SSD network, the accuracy rate of Faster R-CNN with the integrated FPN network is about 0.7 higher. Although the detection time is longer than that of the SSD network, the overall efficiency is still higher than that of the SSD network.

According to the model evaluation, this experiment makes a detailed comparison of the two indexes of accuracy and recall, $F_1$ value, as well as MIoU and detection time (single image). The comparative results of the deep network model are shown in Table 3. It can be concluded from Table 3 that the Faster R-CNN deep network model based on the ResNext-101 network shows better detection performance, while the accuracy and recall rate of the YOLOv3 deep network model are both somewhat lower than those of the Faster R-CNN network. The accuracy of the deep network model can only reach about 84% and the recall rate can only reach about 80%, the $F_1$ value is about 8% lower than that of Faster R-CNN and the difference in the value of MIoU can reach about 9%, although it has a shorter detection time. Hence, the advantages of the Faster R-CNN network are greater.

The Faster R-CNN network model integrated with FPN was analysed via a confusion matrix (Figure 13). The Faster R-CNN network model integrated with FPN was used for the detection of weeds and plant seedlings. The light-coloured areas show very low FN and PN values, most of which are 0 and a few are 1–4. The dark areas show very high values of TP and TN, which shows that the deep learning network predicts a high proportion of correct results; hence, it has high accuracy.

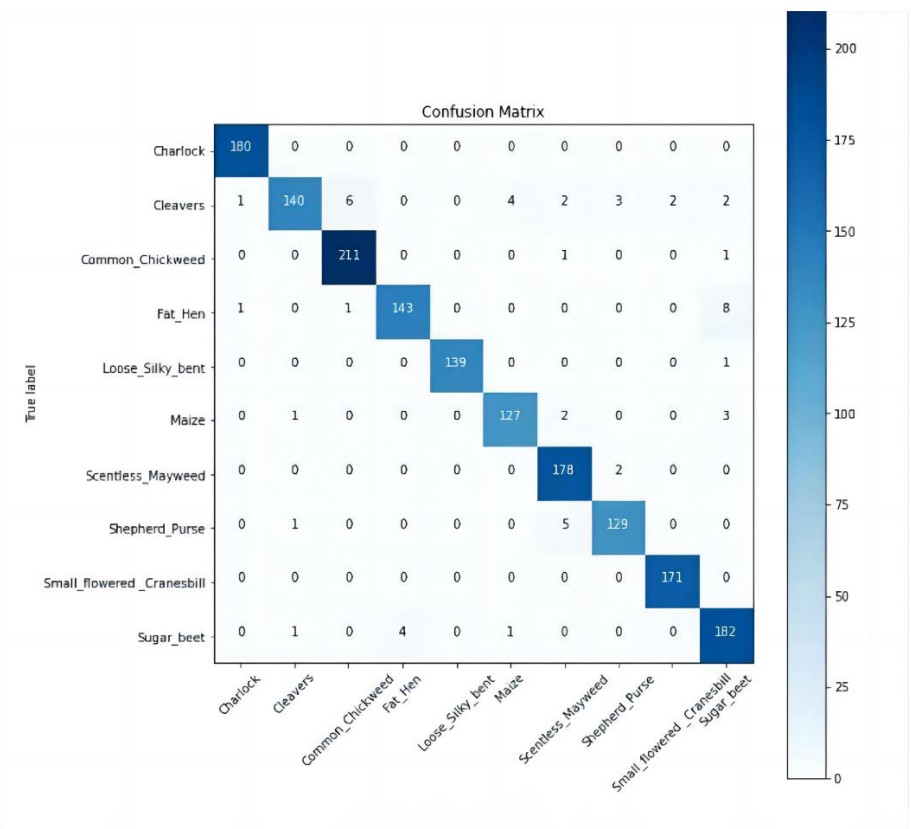

**Figure 13.** Confusion matrix diagram of identification results.

These model training results show the advantages and disadvantages of the two feature extraction networks (ResNet-50 and ResNeXt-101) in weed recognition. It is found that, compared with the Faster R-CNN deep network model, the YOLOv3 and SSD deep

network models have slightly inferior weed recognition accuracy. The Faster R-CNN network model obtains much better accuracy by integration with the FPN network, while the improved ResNeXt feature extraction network has higher computational efficiency. The Faster R-CNN network model integrates the FPN network into the feature extraction network and adopts the process of bottom-to-top, bottom-up, and horizontal connection to realize the simultaneous use of high-resolution low-level features and high-level features. Integrating the features of these different layers to achieve the prediction effect can greatly improve the detection accuracy. The selected ResNeXt-101 feature extraction network achieves more accurate target recognition by using a convolution operation with multiple paths. The experimental results show that the Faster R-CNN deep network model based on the ResNeXt-101 feature extraction network has obvious advantages for weed recognition in field images after being integrated with the FPN network.

## 6. Conclusions

This paper used corn beet and other crop seedling and weed image data to build a Faster R-CNN deep network model based on the ResNeXt-101 feature extraction network and the Tensorflow 2.0 (Developed by Google, Inc. in California, USA) deep learning framework. The FPN network and improved ResNeXt network were applied to the identification of weeds. The actual effects of the three feature extraction networks were compared and analysed, and the key parameters of the Faster R-CNN were optimized to make it more suitable for weed identification in field images with complex backgrounds.

The experimental results show that the Faster R-CNN deep network model obtains improved recognition accuracy by using the ResNeXt feature extraction network and incorporating the FPN network. It has obvious advantages compared with the ResNet feature extraction network in achieving rapid and effective target recognition, and demonstrates the great efficiency of deep learning methods in this field.

**Author Contributions:** Conceptualization, Y.M. and R.F.; methodology, Y.M.; software, R.F.; validation, R.N., J.L. and T.L. (Tianye Luo); formal analysis, Y.S.; investigation, T.L. (Tonghe Liu); resources, H.G.; data curation, X.L.; writing—original draft preparation, Y.M.; writing—review and editing, S.L. and Y.W.; visualization, Y.B.; supervision, Y.G.; project administration, T.H. All authors have read and agreed to the published version of the manuscript.

**Funding:** This research was funded by Jilin Province Science and Technology Development Plan (focuses on research and development projects), funding number 20200402006NC (http://kjt.jl.gov.cn) accessed on 1 January 2020, Key technology R&D project of Changchun Science and Technology Bureau of Jilin Province, funding number 21ZGN29 (http://kjj.changchun.gov.cn) accessed on 1 November 2021 and Science and Technology Research Project of Jilin Provincial Department of Education, funding number JJKH20220337KJ (http://jyt.jl.gov.cn/) accessed on 1 January 2022.

**Data Availability Statement:** All new research data were presented in this contribution.

**Conflicts of Interest:** The authors declare that they have no conflict of interest.

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
