# Peer review of "A Faster R-CNN-Based Model for the Identification of Weed Seedling"

_agronomy, doi:10.3390/agronomy12112867_

Round 1
Reviewer 1 Report
This paper examined a CNN experiment on the image segmentation of crop fields, which is a reasonable approach and can be considered publication after some modification regarding the performance statistics.
Line 184. mIOU (mean intersection over union) is now a standard indicator for image segmentation. Please add mIOU to Section 4.3.
Tables 2 and 3. Please add mIOU to these tables.
Figure 14. The class names are not distinguishable. Please provide a high-resolution figure.
Author Response
Response to Reviewer 1 Comments
Point 1: Line 184. mIOU (mean intersection over union) is now a standard indicator for image segmentation. Please add mIOU to Section 4.3.
Response 1: The addition and detail of MIoU has been completed in section 4.3.
Point 2: Tables 2 and 3. Please add mIOU to these tables.
Response 2: MIoU has been added to Tables 1, 2, 3 and is described.
Point 3: Figure 14. The class names are not distinguishable. Please provide a high-resolution figure.
Response 3: Figure14 has been modified to provide a high-resolution image.
Attachments are modified documents and traces of modifications remain.
Reviewer 2 Report
The authors have proposed a faster R-CNN model for weed identification in field images. The CNN model is well-justified and presented.
My main concern regards the image dataset used, which in my opinion shows strong limitations to the work.
1. The weeds shown are always isolated (i.e. one per image), and the background is filled with marbles. This is not common in crop fields, where the soil will usually be different in color, texture, and the weeds will be spread in the image.
2. Although we know it is difficult to put together a dataset more realistic, the authors could:
2.1 maybe modify the title, since "...crop-field images..." are different (as explained) than the ones used;
2.2 write a section spelling out clearly those limitations regarding the dataset used;
This should improve the paper without putting out a reader with stronger expectations about the applicability of the model.
Author Response
Response to Reviewer 2 Comments
Point 1: The weeds shown are always isolated (i.e. one per image), and the background is filled with marbles. This is not common in crop fields, where the soil will usually be different in color, texture, and the weeds will be spread in the image.
Response 1: This problem has been explained in the article, expressing the disadvantages of using this data set. Clear instructions are given for the non-use of a single image in the dataset in the field.
Point 2: Maybe modify the title, since "...crop-field images..." are different (as explained) than the ones used.
Response 2: The title has been modified to read:”A Faster R-CNN-based model for the identification of weeds seedling”.
Point 3: Write a section spelling out clearly those limitations regarding the dataset used.
Response 3: The shortcomings and shortcomings of the dataset have been described in detail.In the article, it is mentioned that there will be certain restrictions due to the fact that problems such as a single photo cannot be widely used.
Attachments are modified documents and traces of modifications remain.
Round 2
Reviewer 2 Report
It would be interesting if the authors provide data and code so others could work and increase citations on similar problems.